# Prevention of acute rejection after rescue with Belatacept by association of low-dose Tacrolimus maintenance in medically complex kidney transplant recipients with early or late graft dysfunction

Ester Gallo[1☯], Isabella Abbasciano[1☯], Silvia Mingozzi[1], Antonio Lavacca[1], Roberto Presta[1], Stefania Bruno[1], Ilaria Deambrosis[1], Antonella Barreca[2], Renato Romagnoli[3], Alberto Mella[1], Fabrizio Fop[1], Luigi Biancone[1]*

1 Renal Transplant Center "A. Vercellone", Nephrology, Department of Medical Sciences, Dialysis and Renal Transplant Division, "Città della Salute e della Scienza" Hospital, University of Turin, Turin, Italy,
2 Department of Medical Sciences, Division of Pathology, "Città della Salute e della Scienza" Hospital, University of Turin, Turin, Italy, 3 Liver Transplant Unit, General Surgery 2U, "Città della Salute e della Scienza" Hospital, University of Turin, Turin, Italy

☯ These authors contributed equally to this work.
* luigi.biancone@unito.it

## Abstract

### Background

Increased acute rejection risk in rescue protocols with Belatacept may limit its use particularly in medically complex patients where preexisting increased risk of rejection couples with CNI toxicity.

### Methods

Retrospective analysis was performed in 19 KTs shifted to a Belatacept-based immunosuppression with low-dose Tacrolimus (2–3 ng/mL) after evidence of allograft disfunction, including patients with primary non-function (PNF), chronic-active antibody-mediated rejection (cAMR), history of previous KTs and/or other concomitant transplants (liver, pancreas). Evaluation of $CD28^+$ $CD4^+$ effector memory T cell ($T_{EM}$) before conversion was performed in 10/19.

### Results

Kidney function significantly improved (median eGFR 16.5 ml/min/1.73m$^2$ before vs 25 ml/min after; p = 0.001) at a median time after conversion of 12.5 months (9.1–17.8). Overall graft and patient survival were 89.5% and 100% respectively. Definitive weaning from dialysis in 5/5 KTs with PNF was observed, whereas 7/8 patients lost their graft within first year in a control group. eGFR significantly ameliorated in re-trasplants (p = 0.001) and stabilized in KTs with other organ transplants or cAMR. No acute rejection episodes occurred, despite the significant risk suggested by high frequency of $CD28^+$ $CD4^+$ $T_{EM}$ in most patients. Opportunistic infections were limited and most common in early vs late-converted.

**Data Availability Statement:** All relevant data are within the paper and its Supporting Information files.

**Funding:** The authors received no specific funding for this work.

**Competing interests:** The authors have declared that no competing interests exist.

## Conclusions

Rescue association of Belatacept with low-dose Tacrolimus in medically complex KTs is a feasible option that allows prevention of acute rejection and amelioration of graft function.

## Introduction

Belatacept, a selective costimulation blocker consisting of soluble CTLA4/IgG fusion protein, prevents T cell CD28 signaling by efficiently binding with its ligands CD80 and CD86 expressed by antigen-expressing cells (APCs) [1–3]. A long-term trial has shown an improvement of graft survival in kidney transplanted patients in comparison to cyclosporine [2, 4]. Improved graft function was observed also in comparison to Tacrolimus (TAC) maintenance [5].

However, an increased incidence of acute rejection (AR) [2, 4, 6–8] in patients treated with Belatacept was observed, mainly in free-calcineurin inhibitors (CNI) regimen [9], and raised the concern of its use in patients with moderate or high immunologic risk. AR occurs very early, 82% within three months from conversion [2]. Recently, Adams et al [5] contained the incidence of AR in patients started on Belatacept from the beginning of the transplant by transiently combining TAC to Belatacept. In order to obtain an acceptable rejection rate (about 16%) TAC should be tapered slowly in 9 months after KT [5].

Based on its characteristics, Belatacept is now been mainly adopted as rescue therapy in case of CNI-induced nephrotoxicity or graft function impairment, especially in marginal kidneys recipients [10, 11]. Both early and late conversion were explored [10, 12, 13]. Switch to Belatacept within three months after KT demonstrated better results in terms of estimated glomerular filtration rate (eGFR) increasing [10]. However, also in this setting, AR occurs (8.2% in Retrospective Multicenter European Study [10], 4% and 11.4% in Le Meur et al's study [14] and in Brakemeier et al's study [15] respectively). The AR rate reaches 25% in Perez-Saez et al [16] probably due to the inclusion of patients at high immunological risk, even if these data are not confirmed by Gupta et al [17].

In both ab-initio and rescue protocols the majority of AR are classified as T-cell mediated (TCMR) [18] with good response to steroids; nonetheless, some patients need a second-line treatment with anti-lymphocyte polyclonal antibodies, and a few number also experiences antibody-mediated rejection (AMR) and graft loss [10]. Moreover, also if the episode was successfully treated, all AR-related therapies are associated with increased morbidity and mortality especially due to infectious cause, with higher risk in old and frail subjects [10, 19].

In the present study, we analyze our experience with the adoption of Belatacept-based immunosuppression in association with low-dose Tacrolimus (2–3 ng/mL) in a specific population of KTs at high immunological risk with a high medically complex profile (i.e. combined transplants). The rational of this protocol is to combine the Belatacept positive effects with a reduced CNI exposure for minimizing the risk of AR.

## Materials and methods

### Study design

We performed a retrospective analysis, including 19 adult KT recipients. Belatacept was associated to maintenance immunosuppressive therapy between May 2017 and August 2019.

Patients were converted in case of a) early allograft disfunction, intended as primary non function (PNF) (dialysis dependence or creatinine clearance <20 ml/min after three months from KT) or persistent graft disfunction (after the third month and within 9 months post KT) or

b) late allograft disfunction [suboptimal kidney function with histological diagnosis of chronic antibody mediated rejection (cAMR) and/or interstitial fibrosis-tubular atrophy (IF-TA)].

Exclusion criteria for Belatacept association were: Epstein Barr virus (EBV) negative serology, pregnancy or breastfeeding, no active contraception for women, acute infections.

All patients were closely monitored for adverse events and severe adverse events, also including serial evaluation of EBV and CMV viral load. CMV prophylaxis was administered post- KT according to donor/recipient serologic status and induction therapy.

The study was performed in adherence with the last version of the Helsinki Declaration and with the Principles of the Declaration of Istanbul on Organ Trafficking and Transplant Tourism. All patients signed an informed consent before switching to Belatacept-based immunosuppressive therapy, including their permission to have data from their medical records used in research. This study is covered by our Ethical Committee (*Comitato Etico Interaziendale A. O.U. Città della Salute e della Scienza di Torino—A.O. Ordine Mauriziano—A.S.L. Città di Torino*) approval, resolution number 1449/2019 on 11/08/2019 ("TGT observational study").

## Betalacept based-immunosuppression protocol

Our immunosuppressive protocol is summarized in Fig 1.

Briefly, Belatacept was administered intravenously at a dose of 5 mg/Kg in 30 minutes on day 1, 15, 29, 43 and 57 with subsequent doses scheduled every 28 days thereafter. During the first two weeks following Belatacept initiation TAC dosage was unchanged. On day 15 it was reduced by 40–50% of the initial dose; after the 3th dose of Belatacept TAC was maintained at trough level 3–5 ng/ml and then 2–3 ng/ml.

Mycophenolate mofetil/mycophenolic acid (MMF/MPA) and prednisone (PN) were also maintained in association with Belatacept and low dose TAC unless clinical conditions required discontinuation.

## Phenotypic analysis of isolated peripheral blood mononuclear cells (PBMC)

Standard extracellular staining was performed on PBMC, using the following fluorophorelabeled antibodies: CD45RA-Pacific Blue (Beckman Coulter), CD28-FITC (Necron Dickinson Pharmigen), CCR7-PE (Biolegend) and CD4-PerCP (Biolegend).

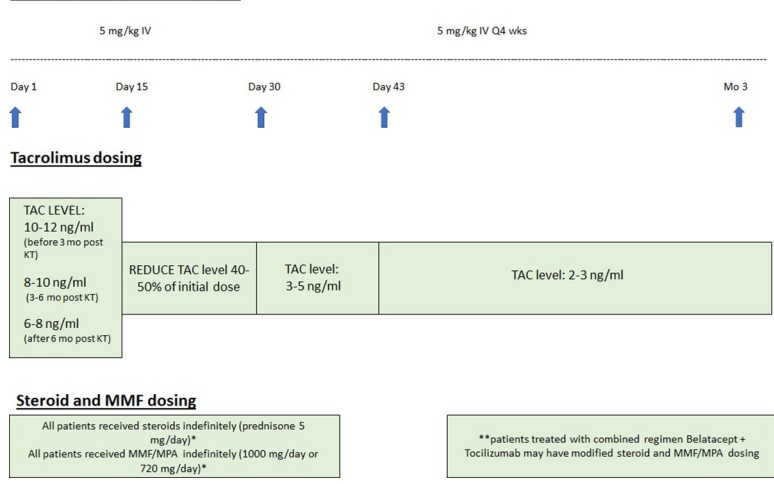

**Fig 1. Belatacept association protocol (timeline).**

CD28$^+$ CD4$^+$ T Effector Memory (T$_{EM}$), previously associated to an increased risk of AR in patients treated with Belatacept [20], were assessed before conversion to belatacept in 10/19.

## Statistical analysis

Continuous variables were described as median and interquartile range (IQR) according to their non-normal distribution. To compare independent groups we used Mann-Whitney test and to compare related variables we used the Wilcoxon signed-rank test.

Categorical variables were presented as fraction and Pearson's or, for small samples, Fisher's exact test was employed to compare groups. Cumulative survival was analyzed by Kaplan-Meier (KM) curves. Significance level for all tests was set at α<0.05.

All statistical analyses were performed using Spss (IBM Corp. Released 2020. IBM SPSS Statistics for Windows, Version 26.0. Armonk, NY: IBM Corp.), including an additional analysis with a historical cohort of KTs with PNF before Belatacept availability in our center or clinical contraindication to its use (negative EBV serology).

# Results

## Patients characteristics and causes of association

A total of 19 KT recipients (13 males, 6 females) were included in the study. Baseline characteristics of our population are summarized in Table 1.

**Table 1. Baseline characteristics of studied population.**

|  | Belatacept-treated patients (n = 19) |
|---|---|
| Age at KT, yrs (IQR) | 50.3 (46.1–64.2) |
| Gender male, n (%) | 13 (68.4) |
| Race |  |
| *Caucasian, n (%)* | 17 (89.5) |
| *Black, n (%)* | 1 (5.3) |
| *Asian, n (%)* | 1 (5.3) |
| Previous KT, n (%) | 7 (36.8) |
| *2$^{nd}$ kidney transplantation, n (%)* | 3 (15.8) |
| *3$^{rd}$ kidney transplantation, n (%)* | 3 (15.8) |
| *5$^{th}$ kidney transplantation, n (%)* | 1 (5.3) |
| KT with other organ transplants, n (%) | 3 (15.8) |
| *Liver transplantation, n (%)* | 1 (5.3) |
| *Pancreas transplantation, n (%)* | 1[a] (5.3) |
| *Liver + pancreas transplantation, n (%)* | 1 (5.3) |
| Pre-emptive KT, n (%) | 2 (10.5) |
| Cumulative time in dialysis, yrs (IQR) | 3.8 (1.9–10.2) |
| Donor characteristics |  |
| *Age, yrs (IQR)* | 63 (51–69) |
| *Deceased donor, n* | 19 (100) |
| *KDRI (IQR)* | 1.43 (1.15–1.66) |
| *KDPI, % (IQR)* | 84 (64–92) |
| Cold ischemia time, hours (IQR) | 14.7 (13.5–16.4) |
| DGF[b], n (%) | 11 (57.9) |
| DGF duration, days (IQR) | 24 (14–41.5) |
| HLA mismatches, n (IQR) | 3 (2.25–4) |

(*Continued*)

**Table 1.** (Continued)

| | Belatacept-treated patients (n = 19) |
|---|---|
| vPRA, % (IQR) | 0 (0–80.5) |
| Rejection before Belatacept, n (%) | 7 (36.8) |
| Acute AMR, n (%) | 1 (5.3) |
| TCMR, n *(%)* | 3 (15.8) |
| cAMR, n *(%)* | 3[c] (15.8) |
| Anti-HLA DSA positive, n *(%)* | 4 (21.0) |
| Time from KT to Belatacept conversion, months (IQR) | 4.2 (1.3–7.4) |
| Conversion <12 months post-KT, n (%) | 15 (78.9) |
| Chronic kidney disease stage at conversion | |
| *Stage 1 or 2 (eGFR>60 mL/min/1.73 m$^2$), n (%)* | 0 (0) |
| *Stage 3 (eGFR 30–60 mL/min/1.73 m$^2$), n (%)* | 1 (5.3) |
| *Stage 4 (eGFR 15–30 mL/min/1.73 m$^2$), n (%)* | 10 (52.6) |
| *Stage 5 (eGFR<15 mL/min/1.73 m$^2$), n (%)* | 8 (42.1) |
| Tacrolimus median level[d] (ng/ml) | |
| *Before conversion* | 8.1 (6.5–9.5) |
| *15 days after conversion* | 7.7 (5.5–9.5) |
| *30 days after conversion* | 5.5 (4.4–7.3) |
| *3 months after conversion* | 3.5 (3–4.7) |
| Follow-up after conversion, months (IQR) | 12.5 (9.1–17.8) |

Data are expressed as median with interquartile range (IQR) when appropriate.

KT: kidney transplant; KDRI: kidney donor risk index; KDPI: kidney donor profile index; DGF: delayed graft function; vPRA: virtual panel reactive antibody; eGFR: estimated glomerular filtration rate; AMR: antibody-mediated rejection; TCMR: T-cell-mediated rejection.

[a] combined pancreas-KT.

[b] intended as dialysis need in the first week after kidney transplantation.

[c] two additional cAMR are diagnosed after conversion.

[d] based on different post-transplant period.

The median age was 50.3 years old (46.1–64.2). All patients received a deceased-donor KT and, according to KDPI index (cut-off 85%) [21], 11/19 were classified as expanded criteria donor (ECD). The median cold ischemia time was 14.7 hours (13.5–16.4).

One out of 19 received a kidney-pancreas combined transplant; 2 patients had another organ transplant before KT (one liver and one liver- pancreas). Seven out of 19 have an history of previous KT, including 3 third and 1 fifth KT. This fifth KT was performed during the treatment with Belatacept, administered as rescue therapy for the failing fourth graft.

Seven patients experienced rejection before Belatacept association: three were diagnosed as acute T-cell mediated rejection (aTMR), one as acute antibody mediated rejection (aAMR), and three as chronic antibody mediated rejection (cAMR). All cAMR KTs started Tocilizumab (TCZ; 8 mg/kg/monthly) before conversion at a median time of 6.7 months (min-max 2.1–9.6).

HLA-DSAs, mostly anti-class II, were already detected in 4 out of 19 patients before the start of Belatacept.

Reason for Belatacept association was rescue therapy in early or late allograft disfunction (median time after KT 4.2 months, 1.3–7.4). Early association was performed in 15 patients (79%, 11/15 with PNF and 4/15 with persistent renal functional impairment). The remaining 4 KTs were late associations with significant IF-TA on kidney biopsies (2/4 were also treated with TCZ for cAMR).

**Table 2. Immunosuppressive medication before and after conversion.**

|  | Belatacept-treated patients (n = 19) |
|---|---|
| Induction at transplants |  |
| *Anti-interleukin 2 receptor blocker, n (%)* | 7 (36.8) |
| *Rabbit anti-thymocyte globulin, n (%)* | 12 (63.2) |
| Maintenance therapy before conversion |  |
| *Tacrolimus, n (%)* | 19 (100) |
| Mycophenolate mofetil, *n (%)* | 18 (94.7) |
| *Azathioprine, n (%)* | 1 (5.3) |
| *Steroids, n (%)* | 19 (100) |
| *Tocilizumab, n (%)* | 3 (15.8) |
| Immunosuppressive therapy after conversion[a] |  |
| *TAC, n (%)* | 19 (100) |
| *MMF, n (%)* | 9 (47.4) |
| *AZA, n (%)* | 1 (5.3) |
| *Steroids, n (%)* | 19 (100) |
| *Tocilizumab, n (%)* | 3[b] (15.8) |

[a] at the end of follow up

[b] 2 patients started Tocilizumab after conversion to belatacept. 2 patients stopped Tocilizumab before the end of follow up

Follow up median time from the association was 12.5 months (9.1–17.8).

Immunosuppressive medications pre- and post- Belatacept were detailed in Table 2.

At transplantation time all patients received induction therapy, consisted of either basiliximab (Simulect; Novartis Pharmaceuticals Corp., East Hanover, NJ) or rabbit anti-thymocyte globulin (rATG; Thymoglobulin; Genzyme, Cambridge, MA) in association with steroids, according to donor (standard or ECD) and recipient characteristics (i.e. immunological risk). Maintenance therapy was composed by TAC, MMF/MPA (18/19) or azathioprine (1/19) and steroids; two out of three patients with other organ transplant already received TAC and MMF/MPA before KT (the remaining one was also treated with steroids). During the observation period after conversion, one patient stopped AZA and 9 MMF/MPA.

In our study population, according to Literature data [22, 23], we used TCZ in patients (5/19) with histological diagnosis of cAMR. In 3 cases, it was associated before and in 2 cases after Belatacept starting. During the follow-up 2 patients suspended TCZ, one because of an episode of severe sepsis from cholangitis secondary to biliary obstruction of the transplanted liver and one for functional deterioration and poor patient compliance.

Eighteen out of 19 patients received a kidney biopsy at a median time before conversion of 3,02 (0.82–6.55) months (Banff scores are summarized in Table 3).

## Renal function, graft survival and acute rejection rate

Fig 2 shows improvements in kidney function after Belatacept association both in the overall (a) and in early (dialysis dependent and not) vs late rescue therapy (b).

A significant estimated glomerular filtration rate (eGFR, CKD-EPI formula) amelioration after belatacept association was observed at the end of f/up (median improvement of 8.5 ml/min/1.73m$^2$; p = 0.001).

Stratifying the population according to shifting time, optimal results were obtained in early-converted KTs (15/19), both in dialysis (HD) and non-dialysis dependent patients. We

**Table 3. Kidney biopsy assessment before conversion (available in 18/19 KTs).**

|  | Patients, n (%) | Median BANFF lesion score[a] (IQR)[b] |
|---|---|---|
| Tubulitis (t) | 3 (16.6) | 2 (min 1; max 3)[b] |
| Intimal arteritis (v) | 1 (5.5) | 3 |
| Glomerulitis (g) | 3 (16.6) | 2 (min 1; max 3)[b] |
| Peritubular capillaritis (ptc) | 4 (22.2) | 1 (1–2.5) |
| C4d positive staining | 4 (22.2) | 2 (1–3) |
| Interstitial fibrosis (ci) | 12 (66.6) | 2 (1–2) |
| Tubular atrophy (ct) | 12 (66.6) | 1 (1–2) |
| Vascular fibrous intimal thickening (cv) | 15 (83.3) | 1 (1–1) |
| Transplant glomerulopathy (cg) | 3 (16.6) | 3 (min 2; max 3)[b] |
| Arteriolar hyalinosis (ah) | 11 (61.1) | 1 (1–1) |
| Interstitial fibrosis-tubular atrophy (IF-TA) | 12 (66.6) | 1 (1–1.75) |

[a]according to BANFF lesion score (0–3)

[b]IQR is not available if n<4, so minimum and maximum values are reported.

underline that HD dependent subgroup with PNF stopped renal replacement therapy after belatacept start, reaching significant improvement of eGFR (30 ml/min/1.73m$^2$, 25–35). A significant difference was clearly noted comparing these patients to an internal cohort of KTs with PNF, similar characteristics and a different immunosuppressive protocol [reduction of TAC level and/or mTOR-inhibitors (mTORi) introduction] (Table 4 and Fig 3). In these eight patients no stable recovery of renal function was reached at a similar f/up (p<0.001) and 7/8 patients lost definitely their graft.

The 10/15 early converted non-HD dependent KTs also showed a significant eGFR improvement (from 17.3 to 23.6 ml/min/1.73m$^2$; p = 0.005).

In late conversion group (4/19) eGFR stabilized without significant increase (17.5 ml/min1.73m$^2$ vs 14.8 ml/min/1.73m$^2$); this result may be related to the fact that 2/4 patients didn't benefit of Belatacept with progressive eGFR decline, needing HD restart.

Fig 4 summarized overall graft survival. No AR episode was reported, according to clinical and laboratory data, also in patients with preexisting DSA (4/19). Moreover, DSA disappeared in all, except one who demonstrated significant reduction of MFI (from 19600 to 7200). DSA development after Belatacept association was not observed in all patients during the follow-up.

Cortes-Cerisuelo et al [20] demonstrated that higher frequencies of CD28$^+$ CD4$^+$ T$_{EM}$ (cut-off value > 83,6%; sensitive 80%-specificity 100%) may predict AR. We analyzed CD28$^+$ expression within CD4$^+$ T$_{EM}$ cells compartment, collecting samples from 10 patients before conversion. In our population, 8 patients showed a percentage of CD28$^+$CD4$^+$ T$_{EM}$ >83,6% but none experienced acute rejection episodes.

## Additional analysis in high medically complex subgroups

Renal function modifications after association of belatacept was analyzed in specific subgroups of high complex patients, such as those with diagnosis of cAMR concomitantly treated with TCZ (5/19), kidney re-transplants (7/19) or other organ transplants besides KT (3/19) (Fig 5).

In cAMR group only one patient lost its graft, with eGFR stabilization in 4/5 (eGFR 18.9 ml/min/1.73m$^2$ pre vs 17.1 ml/min/1.73m$^2$ post association).

Among kidney re-transplants we observed significant amelioration of renal function (12.7 vs 25 ml/min/1.73m$^2$, p = 0.028) at the end of follow-up. In one patient bearing a fourth kidney transplant in end-stage renal disease, Belatacept was able to bridge to a pre-emptive fifth

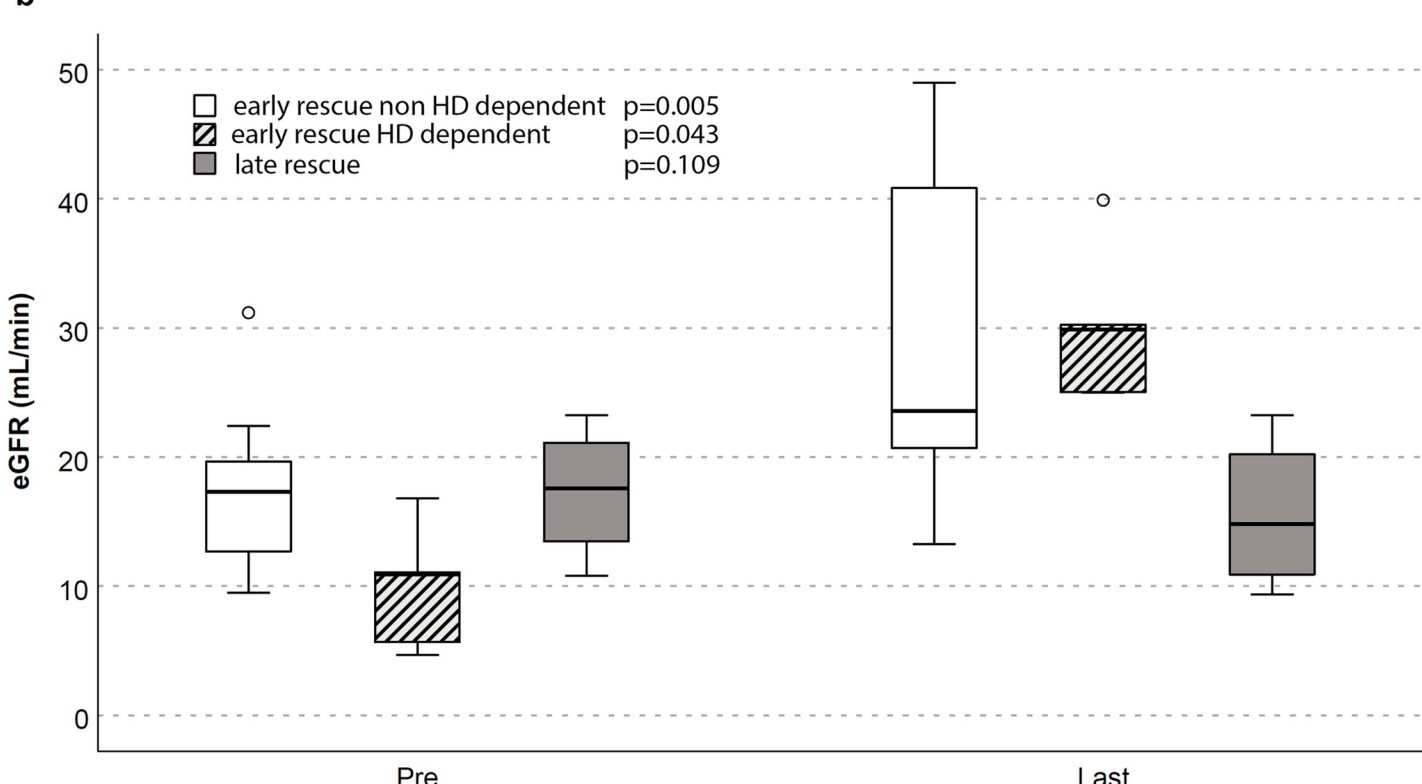

**Fig 2.** eGFR (CKD-EPI formula) before Belatacept association (pre) and at last follow-up (last) in overall population (panel a) and early- or late-converted patients (panel b).

**Table 4. Comparison of baseline characteristics between Belatacept-treated and historical cohort of KTs with dialysis-dependent primary-non function (PNF).**

|  | Belatacept-treated PNF (n = 5) | Historical cohort PNF (n = 8) |
|---|---|---|
| Age, years (IQR) | 45.3 (34.1–66) | 61 (48–71.25) |
| Male gender male, n (%) | 2 (40) | 4 (50) |
| vPRA, % (IQR) | 0 (0–49.5) | 41 (0–88.75) |
| Donor age, years (IQR) | 52 (44–65) | 70.5 (62.25–74.25) |
| KDPI, % (IQR) | 80 (59.5–88) | 93% (78.25–96.5) |
| Previous KT, n (%) | 3 (60) | 3 (37.5) |
| Combined transplant (pancreas-kidney), n (%) | 1 (20) | 1(12.5) |
| DGF duration, days (IQR) | 48 (20–92.5) | 14 [a] (7.5–56) |
| HD dependent patients at 1-year post-KT, n (%) | 0 (0) | 7[b] (87.5) |
| mTORi therapy adoption, n (%) | 0 (0) | 2 (25) |
| Histology at the time of PNF |  |  |
| *Acute tubular necrosis, n (%)* | 5 (100) | 5 (62.5) |
| *Acute rejection, n (%)* | 1 (20) | 2 (25) |
| *CNI toxicity, n (%)* | 3 (60) | 2 (25) |
| Adverse events at 1- year post-KT, patients n (%) | 2 (40) | 4 (50) |
| *Pyelonephritis, patients n (%)* | 1(20) | 4 (50) |
| *CMV replication, patients n (%)* | 0 (0) | 3 (37.5) |
| *Herpes Zoster disease, patients n (%)* | 0 (0) | 1 (12.5) |
| *Herpes Simplex virus esophagitis, patients n (%)* | 0 (0) | 1 (12.5) |
| *Acute cholecystitis, patients n (%)* | 0 (0) | 1 (12.5) |
| *EBV replication, patients n (%)* | 0 (0) | 1 (12.5) |
| *Acute bacterial pneumonia, patients n (%)* | 0 (0) | 1 (12.5) |
| *Acute interstitial pneumonia, patients n (%)* | 0 (0) | 1 (12.5) |
| *Sepsis of digestive tract origin, patients n (%)* | 1 (20) | 0(0) |

[a]4/7 remain dialysis-dependent; 3/7 partially recovery at a mean time after KT of 5.3 months but definitively restart renal-replacement therapy in <1 year.

[b]1/8 restart dialysis 3 years after KT.

KDPI: kidney donor profile index; HD: hemodialysis; mTORi: mTOR inhibitors; ATN: acute tubular necrosis.

transplant that was successfully performed, nine months later, in the same immunosuppressive regimen.

The low number of patients with other organ transplants does not allow to really demonstrate an eGFR improvement in this subgroup; additionally, 2/3 were late conversion with both severe renal failure and IF-TA before the shift.

## Cardiovascular and metabolic changes

All patients were monitored for systolic and diastolic blood pressure during the observational period. No significant changes in mean systolic and diastolic (vs) blood pressures were observed before conversion and at the end of the follow-up (133±21 and 78±15 mmHg vs 134 ±14 and 76±11 mmHg, respectively). A total of 11 patients (57,9%) received at least one antihypertensive medication at baseline and at last f/up; among them, 3/11 were treated with three or more drugs before conversion vs 4/11 at last evaluation.

Mean cholesterol values were similar during the observation time (167±47 mg/dL at baseline vs 166±49 at the end of the follow-up). Despite not significant, a decrease in triglyceride

**Graft survival**

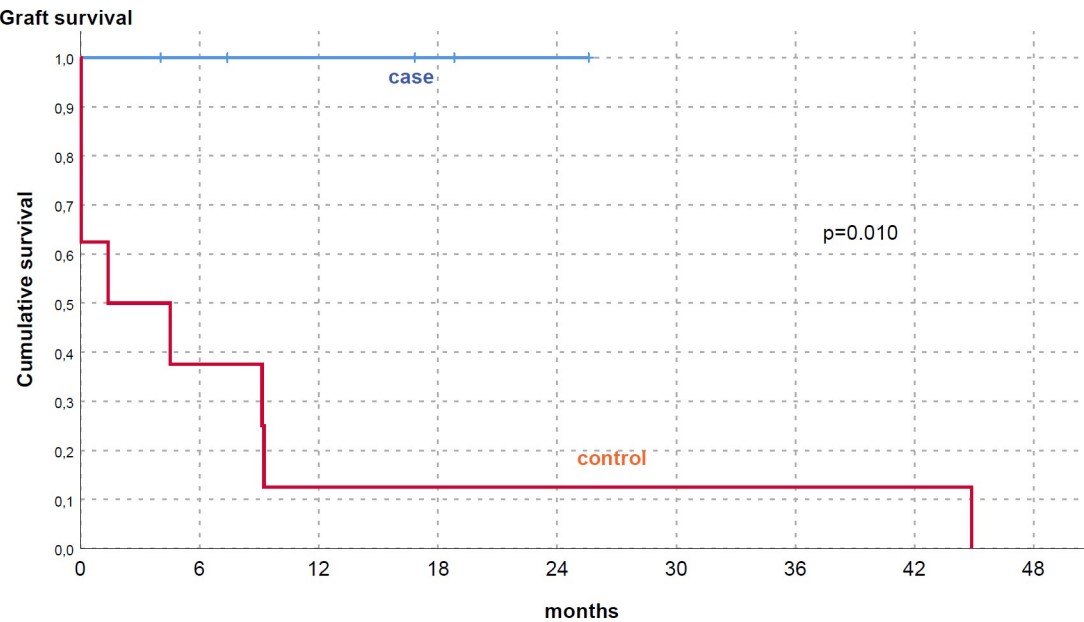

**Fig 3. Cumulative graft survival in Belatacept treated PNF (case) vs historical group of PNF treated with other immunosuppressive medications (control).**

values with a combined increase in HDL cholesterol was observed after conversion (168 ±73 mg/dL and 42± 13 vs 133±38 mg/dL and 49±18 mg/dL, respectively). There were no significant differences in the proportion of patients using lipid-lowering drugs.

In a total of 16 patients (84%) HbA1c was measured. Compared to baseline, HbA1c levels were slightly lower at the end of the follow-up (39±9 mmol/L vs 38±10). Six patients were treated with subcutaneous insulin at baseline, 4/6 after occurrence of new-onset diabetes after transplantation (NODAT); one had a pancreas transplantation before KT needing insulin therapy since the first transplant, and another one discontinued insulin administration after conversion due to improved glycemic control.

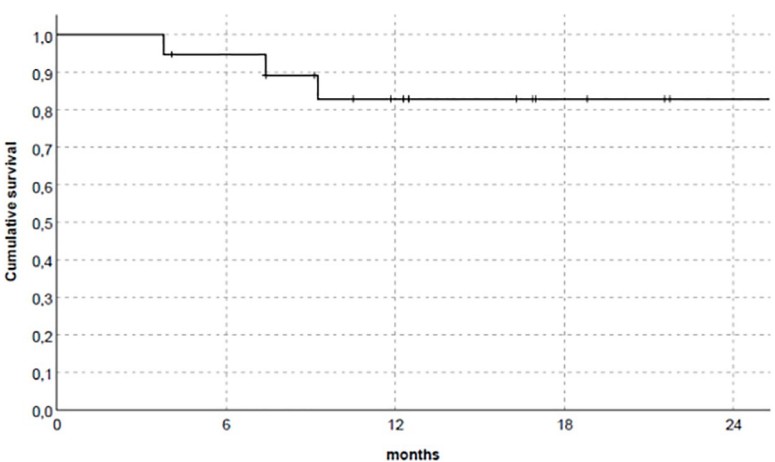

**Fig 4. Overall graft survival in studied population.**

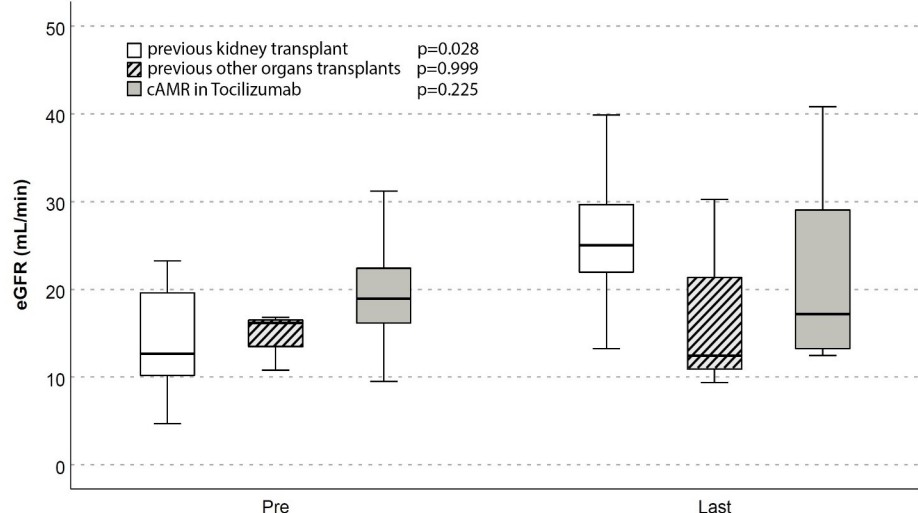

**Fig 5. eGFR (CKD-EPI) before Belatacept association (pre) and at last follow-up (last) in cAMR, re-transplants or patients with other organ transplant.**

## Adverse events

No death or neoplastic complication (including post-transplant lymphoproliferative disease) were recorded during the follow-up. Twelve out of 19 patients experienced viral (including CMV and EBV reactivation) or bacterial infections (incidence 0.062 episode/month of exposure considering a cumulative exposure time of 257 months of Belatacept) (Table 5).

Infection episodes are more common in early (94% of total events) vs late-converted KTs. The only patient in late rescue group experienced acute cholangitis secondary to biliary obstruction of the transplanted liver with severe sepsis, leading to graft failure.

Hospitalization was needed only in 7/19 patients with significant clinical symptoms, and all of them recovery after appropriate therapy without Belatacept interruption (except the one KT

**Table 5. Summary of infectious episodes after conversion.**

|  | Belatacept-treated patients (n = 19) |
|---|---|
| Viral events, n (%) | 8 (42.1) |
| *Epstein-Barr replication, n (%)* | 3 (15.8) |
| *Cytomegalovirus replication, n (%)* | 2 (10.5) |
| *Cytomegalovirus primary infection, n (%)* | 0 (0) |
| *Cytomegalovirus reactivation, n (%)* | 1 (5.3) |
| *Cytomegalovirus disease, n (%)* | 0 (0) |
| *Polyomavirus BK replication, n (%)* | 1 (5.3) |
| *Polyomavirus-associated nephropathy, n (%)* | 0 (0) |
| *Hepatitis B virus replication, n (%)* | 1 (5.3) |
| Bacterial infection, n (%) | 7 (36.8) |
| *Pyelonephritis, n (%)* | 3 (15.8) |
| *Pneumocystisis pneumonia, n (%)* | 1 (5.3) |
| *Sepsis of digestive-tract origin, n (%)* | 1 (5.3) |
| *Cerebral toxoplasmosis, n (%)* | 1 (5.3) |
| *Acute cholangitis, n (%)* | 1 (5.3) |

with cerebral toxoplasmosis who stopped the drug). Pneumocystis pneumoniae occurred in one patient, who ended routinely 6 months post-KT prophylaxis before the event.

## Discussion

Improvement in both graft function and survival for KTs treated with Belacept vs cyclosporine ab-initio is clearly demonstrates in BENEFIT study [1, 2, 24, 25]. Similar results on large registry data are observed comparing Belatacept to TAC [5]. However, in all available protocols AR incidence is not negligible, raising concerns about drug adoption in real-life settings [17, 18, 26, 27]. This issue may be of great importance in high-complexity KTs with combined evidence of CNI toxicity (i.e. patients with previous history of KTs, high PRA and or combined-organ transplants): in these patients an approach to minimize CNI exposure is challenging but imperative, considering that re-transplant may be an unfeasible option in most cases. A theorical use could be also hypothesized in children, despite Belatacept is being studied cautiously in this subgroup owing to concern about the risk of EBV-related PTLD [28].

Rescue conversion to Belatacept was shown to have a 20% risk of rejection [2, 19]. Most of the observed rejection events were cellular rejection and resolved after steroid pulse; however, few of them need treatment with anti-lymphocyte polyclonal antibodies, few were antibody-mediated rejection and some graft losses occurred [18]. In association, even if a rejection event is cured, its treatment may be associated with increased infection morbidity and even mortality, particularly for older patients. An even higher risk of rejection after conversion (25%) has been observed by Perez-Saez et al [16] and has been associated to the higher rate of retransplants and PRA% >30% in their case-series.

In a recent report [29] the avoidance of CNI and steroids in maintenance immunosuppressive regimen with Belatacept, showed a high risk of AR (36%), causing the early interruption of the trial.

Adams et al effectively reduces AR episodes maintaining Belatacept and TAC transiently for 9 months after transplantation, without an increase in recorded infections [5]. The association of CNI with Belatacept in the maintenance immunosuppressive regimen has not been originally conceived in the obvious attempt of overcoming the need of chronic use of CNI. Moreover, in preclinical studies, association of CNI with Belatacept paradoxically lead to increase in rejection rate and reduction of allograft tolerance [30].

In our study we explored the feasibility of the maintenance of TAC at very low doses after association of Belatacept in patients where the rejection risk was higher and/or where the occurrence of rejection with subsequent treatment side effects would have been more dangerous due to the medical complexity of these KTs.

Even in the absence of CNI withdrawal, the drastic reduction in its exposure induces a significant eGFR amelioration, especially in patients with HD-dependent PNF. In this setting, the benefit of conversion was particularly striking by acting as a graft saving protocol when compared to the outcome of a historical cohort.

Alternative strategies to reduce CNI levels without exposing to excessive risk of rejection in the case of prolonged DGF and poor graft function may theoretically involve the use of mTOR inhibitors (mTORi), but these drugs may conversely reduce acute tubular necrosis recovery through their anti-proliferative effect, and are frequently not well-tolerated (30% of drop-out rate due to side-effects) [31, 32].

A portion of the present cohort was composed by patients bearing also a liver or a pancreas transplant, and in one case both. An initial trial comparing Belatacept vs TAC regimen in liver transplantation was stopped early for increased rate of AR, graft loss and death in the Belatacept arm [33, 34]. This is the reason why we maintained low-dose TAC in these patients.

There is no published experience of the use of Belatacept in pancreas transplantation, and therefore this work includes the first report.

We also adopt Belatacept as rescue therapy in a selected group of patients with cAMR. All these patients were contemporary treated with TCZ, a IL-6 receptor-blocker, which previously showed positive effects in cAMR by reducing DSAs and microvascular inflammation (g+ptc score) [22, 23].

Based on our observation of the absence of rejection with combined Belatacept/low-dose TAC, we adopted this regimen together with TCZ as a rescue in patients with poor graft function, cAMR and coexisting histological signs of acute CNI nephrotoxicity. In 4 out of 5 patients this protocol achieved eGFR stability whereas one patient is currently on dialysis after an episode of severe sepsis from a cholangitis secondary to biliary obstruction of the transplanted liver.

Considering the observational nature of the study and the characteristics of the patients, it is difficult to determine if the combination of Belatacept with low-dose TAC increased infections. Particular infectious events, such as cerebral toxoplasmosis or pneumocystis pneumonia, that are usually rare albeit possible in immunosuppressed patients, were observed in one case each.

Indeed, opportunistic infections after full conversion to Belatacept were also recently observed by Bertrand et al [35]. In our work, where low dose TAC is kept, seems consistent with his observation [35]. In the study of Adams et al [5] the combined treatment didn't show difference in serious infection rate when compared to standard TAC-based regimen. In association, in a rescue setting the risk of infection has to be compared to that occurring in case of dialysis reentry, where a marked increase in mortality risk mostly due to infectious events is present [36, 37]. This may be particularly true for patients with a previous transplant of other organs (i.e. liver or pancreas) who are forced to maintain full immunosuppressive therapy.

We are aware of the limits of our study (low numerosity, retrospective design, absence of a control group) and, obviously, randomized controlled trials for rescue protocols are highly needed to set the exact risk/benefit ratio. On the other hand, high medically complex patients who may mostly benefit from this therapeutic approach are frequently excluded from RCT.

In conclusion, in our experience maintenance of low-dose Tacrolimus after rescue conversion to Belatacept in high medically complex KTs is a feasible option that allows prevention of acute rejection, amelioration of graft function without increasing substantially infectious complications.

These data may expand the use of Belatacept, thus conferring new interesting perspectives for its adoption.

## Supporting information

**S1 Data. The raw data elaborated in this study are included in S1 Data [sheet 1: Overall population; sheet 2: Belatacept-treated PNF (cases) vs Historical cohort PNF (controls)].** (XLSX)

## Acknowledgments

The authors thank Drs. M. Scaldaferri and F. Cattel (Pharmaceutical Committee, Città della Salute e della Scienza di Torino, University Hospital) for their expert support.

## Author Contributions

**Conceptualization:** Ester Gallo, Isabella Abbasciano, Silvia Mingozzi, Antonio Lavacca, Roberto Presta, Antonella Barreca, Renato Romagnoli, Alberto Mella, Luigi Biancone.

**Data curation:** Isabella Abbasciano, Silvia Mingozzi, Roberto Presta, Fabrizio Fop.

**Formal analysis:** Silvia Mingozzi, Fabrizio Fop.

**Methodology:** Stefania Bruno, Ilaria Deambrosis, Fabrizio Fop.

**Supervision:** Isabella Abbasciano, Luigi Biancone.

**Validation:** Luigi Biancone.

**Writing – original draft:** Ester Gallo, Isabella Abbasciano, Silvia Mingozzi, Alberto Mella, Luigi Biancone.

**Writing – review & editing:** Ester Gallo.

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
