## [Decision Letter · Decision Letter 0]

5 Aug 2020

PONE-D-20-19734

Prevention of acute rejection after rescue with Belatacept by association of low-dose Tacrolimus maintenance in medically complex kidney transplant recipients with early or late graft dysfunction

PLOS ONE

Dear Dr. Biancone,

Thank you for submitting your manuscript to PLOS ONE. After careful consideration, we feel that it has merit but does not fully meet PLOS ONE’s publication criteria as it currently stands. Therefore, we invite you to submit a revised version of the manuscript that addresses the points raised during the review process.

We look forward to receiving your revised manuscript.

Kind regards,

Paolo Fiorina, MD, PhD

Academic Editor

PLOS ONE

Journal Requirements:

2. In ethics statement in the manuscript and in the online submission form, please provide additional information about the patient records used in your retrospective study. Specifically, please ensure that you have discussed whether all data were fully anonymized before you accessed them and/or whether the IRB or ethics committee waived the requirement for informed consent. If patients provided informed written consent to have data from their medical records used in research, please include this information.

3. Thank you for including your ethics statement:  "HUMAN RESEARCH.

The study was performed in adherence with the last version of the Helsinki Declaration and with the Principles of the Declaration of Istanbul on Organ Trafficking and

Transplant Tourism. All patients signed an informed consent before switching to Belatacept-based immunosuppressive therapy. This study is covered by Ethical

Committee approval, resolution number 1449/2019 on 11/08/2019 ("TGT" observational study)".   

Reviewers' comments:

Reviewer's Responses to Questions

**Comments to the Author**

1. Is the manuscript technically sound, and do the data support the conclusions?

Reviewer #1: Yes

Reviewer #2: Yes

2. Has the statistical analysis been performed appropriately and rigorously? 

Reviewer #1: Yes

Reviewer #2: Yes

3. Have the authors made all data underlying the findings in their manuscript fully available?

Reviewer #1: Yes

Reviewer #2: Yes

4. Is the manuscript presented in an intelligible fashion and written in standard English?

Reviewer #1: Yes

Reviewer #2: Yes

5. Review Comments to the Author

Reviewer #1: The tables are very clear and the test partition is very useful for understanding the study.

For personal interest, the autors can read a work of Vikas et al. (2014), that explain the aspects of kidney transplantation in children.

Reviewer #2: The goal of this study, a suitable maintenance of immunosuppression regimen following kidney transplantations, clearly expressed in the title, is noteworthy. Nevertheless the important purpose of this work is hampered by methodological issues mainly featured by the choiche of the retrospective study method in face of a reduced sample size and without a real control group. Although the study results are consistent with the previous wider studies outcomes, these appear to be powerless. The authors are completely aware of this.

In this study relevant is the researchers interest to analyze medically complex patients otherwise excluded from the more powerfull Randomized Controlled Trials, but the worth efforts to manage highly complex patients deserve an higher patients number to provide more solid informations in this search field.

Minor issue concerns knowing more details about any other previous or concomitant immunosuppressant therapies in those patients received other organ transplants.

Finally it would be valued to obtain insights about metabolic data following belatacept therapy.

6. PLOS authors have the option to publish the peer review history of their article (what does this mean?). If published, this will include your full peer review and any attached files.

Reviewer #1: No

Reviewer #2: No

---

## [Author Response · Author response to Decision Letter 0]

7 Sep 2020

Editorial Review:

1. Response: We have modified our manuscript according to PLOS ONE's style requirements, including those for file naming.

2.Response: Information about patients’ consent are now included in both Material and Methods section and Ethics Statement field of the submission form.

3.Response: The full name of the ethics committee that approved our study is now included in both Material and Methods section and Ethics Statement field of the submission form.

4.Response: We used Preflight Analysis and Conversion Engine (PACE) digital diagnostic tool for all our figures.

Author response to reviewers’ comments:

Reviewer #1

Response: We really thank rev#1 for his/her comments, and we have included the suggested paper (Vikas R. Dharnidharka et al, NEJM 2014) in discussion (Page 18, Line 296-298).

Reviewer #2: 

Response: We appreciate Rev#2 considerations which allow us to improve the quality of our paper. Regarding the limits of our study, as mentioned, we are completely aware of all suggested limitations; however, we proposed some consideration about highly complex patients that, despite the (relative) low number included in our analysis, could be at high importance in this research area, and provided some information which may be useful for both clinicians and researchers.

Regarding immunosuppressive medications in patients with other organs, we added data about previous immunosuppressants in Page 11, Line 180-182.

Insights about metabolic data following belatacept therapy are now included in an appropriate section (Cardiovascular and methabolic changes, Pages 16-17, Line 256-273). Despite the absence of statistical significance, there was a trend in amelioration of some paramenters such as triglycerides. We deeply thank this reviewer to have oriented us to this issue since it can be a matter of further focused studies.

---

## [Decision Letter · Decision Letter 1]

24 Sep 2020

Prevention of acute rejection after rescue with Belatacept by association of low-dose Tacrolimus maintenance in medically complex kidney transplant recipients with early or late graft dysfunction

PONE-D-20-19734R1

Dear Dr. Biancone,

We’re pleased to inform you that your manuscript has been judged scientifically suitable for publication and will be formally accepted for publication once it meets all outstanding technical requirements.

Kind regards,

Paolo Fiorina, MD, PhD

Academic Editor

PLOS ONE

Additional Editor Comments (optional):

Reviewers' comments:

Reviewer's Responses to Questions

**Comments to the Author**

1. If the authors have adequately addressed your comments raised in a previous round of review and you feel that this manuscript is now acceptable for publication, you may indicate that here to bypass the “Comments to the Author” section, enter your conflict of interest statement in the “Confidential to Editor” section, and submit your "Accept" recommendation.

Reviewer #1: All comments have been addressed

2. Is the manuscript technically sound, and do the data support the conclusions?

Reviewer #1: Yes

3. Has the statistical analysis been performed appropriately and rigorously? 

Reviewer #1: Yes

4. Have the authors made all data underlying the findings in their manuscript fully available?

Reviewer #1: Yes

5. Is the manuscript presented in an intelligible fashion and written in standard English?

Reviewer #1: Yes

6. Review Comments to the Author

Reviewer #1: The authors addressed all the previous comments. The manuscript can now be considered for pubblication.

7. PLOS authors have the option to publish the peer review history of their article (what does this mean?). If published, this will include your full peer review and any attached files.

Reviewer #1: No

---

## [Editor Report · Acceptance letter]

5 Oct 2020

PONE-D-20-19734R1 

Prevention of acute rejection after rescue with Belatacept by association of low-dose Tacrolimus maintenance in medically complex kidney transplant recipients with early or late graft dysfunction 

Dear Dr. Biancone:

I'm pleased to inform you that your manuscript has been deemed suitable for publication in PLOS ONE. Congratulations! Your manuscript is now with our production department. 

Kind regards, 

on behalf of

Dr. Paolo Fiorina 

Academic Editor

PLOS ONE